# MFH-NAS: A Hybrid Neural Architecture Search Framework for Multimodal Fusion Object Detection

**Quanwei Gao**[1] **Shuqi Zhao**[1] **Ruyu Wang**[2] **Shuyin Zhang**[1] **Cong Liu**[2] **Zirui Luo**[2]

## Abstract

Multimodal fusion object detection faces a substantial modality gap at the same backbone stage. This makes predefined stage-aligned fusion insufficient for cross-stage interactions. We propose MFH-NAS, a hybrid neural architecture search framework that automatically discovers fusion architectures to better leverage cross-modal complementarity. MFH-NAS searches both local fusion primitives and stage-level fusion connectivity. It targets fusion operator design and fusion stage selection. It couples differentiable search with evolutionary search. Differentiable search learns architecture parameters for local fusion primitives. Evolutionary search explores global fusion topologies, including stage selection and cross-stage connection patterns. The joint search balances exploitation and exploration and mitigates premature convergence. It yields fusion structures that strengthen cross-stage interactions. We evaluate MFH-NAS on three public benchmarks, LLVIP, RGBT-Tiny, and M3FD. MFH-NAS consistently outperforms handcrafted fusion-stage designs and prior stage-searching NAS baselines, improving mAP@0.5 from 85.3% to 88.2% over strong fixed-stage fusion methods and delivering gains across all benchmarks. The code is available at https://github.com/someboy0/MFH-NAS.

## 1. Introduction

Multimodal fusion object detection integrates complementary information acquired from heterogeneous sensors (e.g., visible and infrared), often improves detection performance (Azeem et al., 2024; Yuan et al., 2025; Peng et al., 2025; Li

[1]College of Information Engineering, Northwest A&F University, Yangling, China [2]College of Science, Northwest A&F University, Yangling, China. Correspondence to: Shuyin Zhang <zhangsy217@nwafu.edu.cn>.

*Proceedings of the 43rd International Conference on Machine Learning*, Seoul, South Korea. PMLR 306, 2026. Copyright 2026 by the author(s).

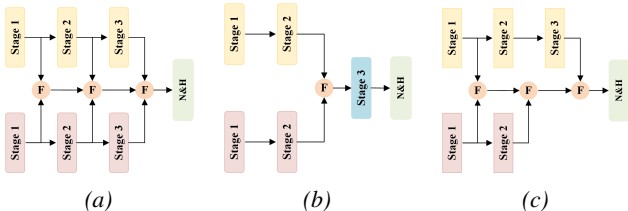

*Figure 1.* Three fusion architectures. (a) and (b) are symmetric fusion architectures, where (a) performs fusion at every stage of feature extraction, while (b) applies fusion only at the intermediate stages of feature extraction; (c) represents an asymmetric fusion architecture that integrates features from different layers across modalities. Yellow, pink, and blue blocks denote visible, infrared, and fused feature processing stages, respectively. F indicates the fusion operation, and N&H represent the neck and detection head.

et al., 2023; Zhao et al., 2023; Zhu et al., 2023; Cao et al., 2023; Zhao et al., 2024a; Liu et al., 2024). The core of this technique lies in the design of fusion modules and the selection of fusion stages.

However, most existing multimodal fusion approaches primarily focus on the design of fusion modules while overlooking the exploration of fusion architectures. As illustrated in Figure 1, the majority of existing multimodal fusion methods adopt predefined, stage-aligned fusion architectures, where features from different modalities are aligned and fused at a specific semantic level (Figure 1 $(a), (b)$). Such designs not only lack a systematic exploration of fusion stages and the number of fusion stages, but also restrict fusion interactions to corresponding backbone stages, making it difficult to fully exploit the semantic complementarity of different modalities across hierarchical levels.

In contrast, non-aligned fusion architectures (Figure 1 $(c)$) establish cross-modal interaction pathways across different stages, enabling more comprehensive cross-stage information flow and semantic complementarity, thereby enhancing representation capability and detection robustness in complex scenarios. Nevertheless, non-aligned fusion architectures exhibit highly combinatorial complexity in terms of stage connections and interaction paths, leading to an exponentially growing architecture space. As a result, manually designed solutions struggle to achieve both efficiency and

optimality.

To address this issue, recent studies have introduced neural architecture search (NAS) to enable the automated design of fusion architectures (Pérez-Rúa et al., 2019; Yu et al., 2020; Lv et al., 2024), allowing more effective exploration of optimal fusion structures under acceptable computational budgets. However, existing NAS-based methods for multimodal fusion architecture design face a critical trade-off between search efficiency and performance. Gradient-based differentiable search methods (e.g., DARTS and its variants) achieve high search efficiency, but due to catastrophic bias during optimization, they are prone to getting trapped in local optima when applied to complex multimodal fusion search spaces (Liu et al., 2018; Hu et al., 2024; Lv et al., 2024; Xu et al., 2019; Zela et al., 2019; Chu et al., 2020; Chen et al., 2020; Han et al., 2023; Mousavi et al., 2023; Miao et al., 2022; Wu et al., 2023; Feng & Wang, 2024). In contrast, evolutionary search algorithms offer stronger global exploration capability, but their high computational cost and slow convergence make them difficult to scale within complex object detection frameworks (Real et al., 2019; Lu et al., 2019; 2020; Fan & Wang, 2023; Liang et al., 2024a; Booysen & Bosman, 2024; Xu & Ma, 2023; Zhao et al., 2024b; Zou et al., 2024).

To overcome the above challenges, we propose MFH-NAS, an improved neural architecture search framework for automatically discovering multimodal fusion architectures for object detection. The core idea of MFH-NAS is to integrate gradient-based optimization and evolutionary search through a dual-level collaborative optimization mechanism. At the microscopic level, MFH-NAS exploits the efficiency of differentiable search to rapidly optimize the weights of fusion operators, enabling efficient identification of high-performing fusion modules and optimal fusion stages. At the macroscopic level, an evolutionary algorithm is employed to extensively explore the global search space, effectively escaping local optima and ultimately achieving better global exploration. The main contributions of this work are summarized as follows:

- We design a Gated Fusion Strategy (GFS) Selector, which is capable of adaptively determining whether multimodal features at the same stage should be fused and selecting appropriate fusion strategies.

- We propose a hybrid neural architecture search framework that deeply integrates the fast convergence of differentiable search with the global robustness of evolutionary algorithms.

- We obtain strong improvements for multimodal object detection. Experiments on LLVIP, RGBT-Tiny and M3FD show that MFH-NAS outperforms both handcrafted fusion-stage designs and existing stage-searching NAS baselines.

## 2. Related Work

### 2.1. Multimodal Fusion Object Detection.

Multimodal object detection relies on effective feature fusion to exploit complementary cues across modalities, making fusion-module design a central research focus. Guo et al. (2024) proposed the DAMSDet method, which introduces a dynamic adaptive multispectral detection transformer that combines competitive query selection with adaptive feature fusion mechanisms to optimize information fusion across different modalities. Similarly, Yuan et al. (2024) proposed the UniRGB-IR framework, which achieves efficient visible-infrared semantic task fusion through adapter-based tuning. Wang et al. (2024b) introduced the CDC-YOLOFusion method, which, based on the YOLO detection framework, employs a cross-scale dynamic convolution module to achieve multi-layer feature adaptive fusion for visible and infrared images. Tang et al. (2024) introduced the ITFuse method, which incorporates an interactive transformer architecture that effectively captures complex inter-modal relationships through an interactive information flow mechanism. In terms of reducing modality inconsistencies, Shen et al. (2024) proposed an iterative cross-attention-guided feature fusion method, which uses a cross-attention mechanism to optimize inter-modal feature fusion. Dong et al. (2025) proposed the Fusion-MAMBA method, which introduces a dynamic learning mechanism and significantly enhances cross-modal object detection performance through multi-level feature fusion and alignment. These methods often optimize fusion at specific stages or employ a single fusion paradigm, but they lack a systematic exploration of fusion stages, limiting the model's adaptability and generalization in complex multimodal scenarios.

### 2.2. Multimodal Fusion Architecture Search.

Multimodal fusion architecture search aims to automatically discover fusion topologies and module configurations under complex multimodal search spaces, reducing the reliance on handcrafted design. Early efforts mainly adopt differentiable NAS to jointly optimize backbones and fusion strategies, such as BM-NAS proposed by Yin et al. (2022). Peng et al. (2025) introduced the Deep Multimodal NAS method, which jointly searches for network structures and fusion units, taking task and hardware constraints into account, thereby advancing the automated optimization of multimodal network architectures. Palladin et al. (2024) were the first to apply NAS to monocular RGB semantic scene completion tasks, proposing a lightweight fusion architecture. Wang et al. (2024a) introduced a multimodal detection architecture search method for LiDAR, RGB, and

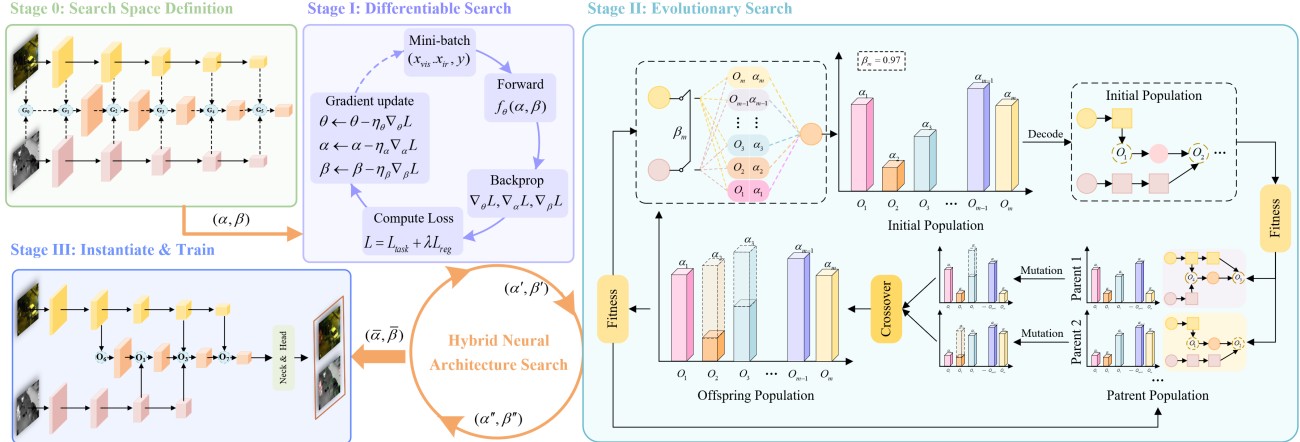

Figure 2. Overview of MFH-NAS with a four-stage pipeline combining differentiable and evolutionary search.

NIR in robotics and autonomous driving, promoting the application of multimodal fusion architectures in perception systems. However, most of the existing search methods focus on the automation of fusion module design, with limited modeling of multi-level and multi-candidate fusion paths.

## 3. Method

We propose MFH-NAS, a hybrid neural architecture search framework that automatically designs multimodal fusion architectures for object detection. As shown in Figure 2, MFH-NAS adopts a four-stage pipeline. Stage 0, highlighted in light green, defines the fusion-architecture search space and initializes the architectural parameters $\alpha$ and $\beta$, which are then passed to Stage I. Stage I, shown in light purple, performs differentiable search to optimize $\alpha$ and $\beta$ within a supernet for a fixed number of epochs, producing updated parameters $\alpha'$ and $\beta'$. These parameters seed Stage II, shown in light cyan, where an evolutionary algorithm conducts population-based exploration over stage-level connectivity and operator configurations. The best evolved parameters $\alpha''$ and $\beta''$ are fed back to Stage I to refine the differentiable search, forming an alternating optimization loop that repeats for $N$ rounds. After convergence, the final architecture parameters $\bar{\alpha}$ and $\bar{\beta}$ are used in Stage III, highlighted in light blue, to instantiate the discovered architecture and train it to obtain the final detection model.

### 3.1. Fusion Architecture Search Space Definition

As illustrated in the green region of Figure 2, our fusion architecture comprises three parallel branches. Two modality-specific backbones extract stage-wise features from visible and infrared inputs, and a fusion branch integrates multi-level information across stages. The yellow and pink blocks

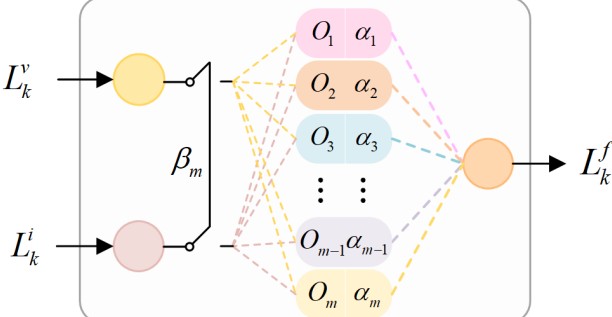

Figure 3. Architecture of the Gated Fusion Strategy (GFS) module.

denote intermediate visible and infrared features, respectively. The fusion branch forms the core of the architecture and is built upon a Gated Fusion Strategy (GFS) selector, denoted by the blue circle labeled $G$, while the orange block represents the fused features.

Based on this architecture, we define a fusion-architecture search space with two searchable units that jointly determine where and how cross-modal interactions are introduced. The first unit is fusion-stage selection. We partition the feature extraction pipeline into $n$ stages and additionally consider the input level, yielding $n + 1$ candidate fusion points. At each candidate point, a binary decision activates or skips feature injection into the fusion branch, resulting in $2^{n+1}$ stage-connection patterns. Excluding the trivial pattern that disables feature injection at all points, the number of valid alternatives is $2^{n+1} - 1$.

The second unit is fusion-method selection, which is implemented within the GFS module. As shown in Figure 3, the GFS provides $m$ candidate symmetric fusion operators $\{O_i\}_{i=1}^{m}$ and introduces learnable architecture parameters

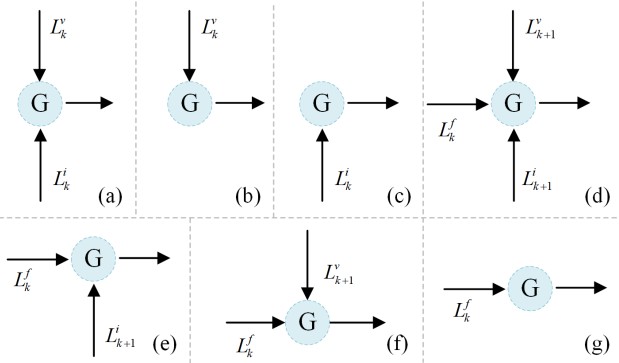

*Figure 4.* Seven GFS operation modes induced by available inputs across stages.

$\boldsymbol{\alpha} = \{\alpha_i\}_{i=1}^{m}$ and $\beta$. The softmax-normalized $\boldsymbol{\alpha}$ weights candidate operators to form a differentiable mixture, while the sigmoid-activated gate $\beta$ controls whether the GFS participates at the current stage.

To characterize the resulting structural variants, Figure 4 enumerates seven operation modes induced by different input availability. Suppose fusion is first enabled at stage $k$. At this stage, the fusion branch either applies one of the $m$ fusion operators to $(L_k^v, L_k^i)$ (Figure 4 (a)) or bypasses fusion by directly forwarding $L_k^v$ or $L_k^i$ (Figure 4 (b)–(c)), yielding $m+2$ alternatives. For any subsequent stage $k+1$, the module additionally receives the previous fused representation $L_k^f$, leading to $m^2$ two-step fusion choices (Figure 4 (d)), $2m$ single-branch fusion choices (Figure 4 (e)–(f)), and one skip choice (Figure 4 (g)). Consequently, stage $k+1$ admits $m^2 + 2m + 1$ fusion-mode choices.

With $n$ candidate stages and $m$ candidate fusion operators, the resulting search space size is

$$S = \sum_{k=1}^{n}(m+2)\,(m^2 + 2m + 1)^{n-k}. \qquad (1)$$

In our setting, we consider $n = 6$ intermediate feature-extraction stages and $m = 8$ candidate fusion operators, which yields $S = 3.53 \times 10^{10}$ possible architectures. This combinatorial growth motivates an efficient search strategy, which we describe next.

It is evident that as the number of fusion methods and stages increases, the network architecture search space expands exponentially, rapidly reaching a scale of trillions. Relying on exhaustive search or neural architecture search methods lacking theoretical guidance would incur immense computational costs, making it difficult to complete an effective search within a reasonable time frame, which would severely undermine the practicality and feasibility of the method. Consequently, the design of an efficient search strategy is imperative.

## 3.2. Two-Stage Search Strategy Design

### 3.2.1. DIFFERENTIABLE GRADIENT DESCENT SEARCH STAGE

The first stage employs differentiable NAS (Liu et al., 2018) for rapid optimization. Specifically, each GFS is treated as an independent search unit. As shown in Figure 3 ($f$), there are $m$ optional fusion methods, each corresponding to a learnable weight parameter $\boldsymbol{\alpha}_i$. After performing softmax normalization on $\boldsymbol{\alpha}_i$, the weighting coefficients for each operation are obtained. The final output of the fusion module is expressed as the weighted sum of all candidate operations. Therefore, the output of the GFS module is defined as:

$$f(x_{vis}, x_{ir}) = \sum_{i=1}^{m} \text{softmax}(\boldsymbol{\alpha}_i) \cdot O_i(x_{vis}, x_{ir}), \qquad (2)$$

where $O_i(x_{vis}, x_{ir})$ represents the output of the $i$-th fusion method applied to the multimodal input features $x_{vis}$ and $x_{ir}$, and $\boldsymbol{\alpha}_i$ denotes the learnable weight for the $i$-th fusion operation. Through (2), the discrete search spaces of the fusion methods is made continuous and becomes differentiable. Furthermore, we introduce a learnable control parameter $\beta$. After passing through the sigmoid activation function, its output value controls whether the GFS module is enabled: a value closer to 1 indicates a higher probability of enabling the module, while a value closer to 0 indicates a higher probability of skipping it. The final fusion output is defined as:

$$g(x_{vis}, x_{ir}) = \text{sigmoid}(\beta) \cdot f(x_{vis}, x_{ir}), \qquad (3)$$

where $\text{sigmoid}(\cdot)$ is the output value of the sigmoid activation function, used to control module activation. The architecture parameters $\alpha$ and $\beta$ serve complementary roles: $\alpha$ weights candidate fusion operators, while $\beta$ controls the activation of each GFS module. In Stage I, both are jointly optimized with the network weights as continuous parameters. In Stage II, they are discretized into operator choices and GFS gate states, and the best evolved architecture feeds its selected operator and active-gate indices back to update the next differentiable round. Finally, we retain GFS modules with $\text{sigmoid}(\beta) > 0.5$ and choose the operator with the largest $\alpha$ for each retained module. By relaxing each operational parameter in the multimodal fusion object detection architecture search space into a continuous space, the gradient descent algorithm be employed to simultaneously optimize the network performance and architecture through joint optimization of network weights and architectural parameters. During the training, the weight of each candidate operation is calculated, and the optimal architecture is selected by minimizing the following loss function:

$$L(\boldsymbol{\theta}, \boldsymbol{\alpha}, \beta) = L_{task}(f_{\boldsymbol{\theta}}(\boldsymbol{\alpha}, \beta)) + \lambda L_{reg}(\boldsymbol{\alpha}, \beta). \qquad (4)$$

Here, $L_{task}$ is the loss function for the target task (such as classification and regression losses in multimodal object detection), the network output $f_{\boldsymbol{\theta}}(\boldsymbol{\alpha}, \beta)$ is defined by the weight vector $\boldsymbol{\theta}$ and the architecture parameters $\boldsymbol{\alpha}$ and $\beta$. The regularization term $L_{reg}(\boldsymbol{\alpha}, \beta)$ prevents an architectural overfitting for the architecture parameters, and $\lambda$ is the weight parameter to controlling architectural complexity. By using the classical gradient descent algorithm, it performs joint optimization of network weight vector and architecture parameters, that is,

$$(\boldsymbol{\theta}^*, \boldsymbol{\alpha}^*, \beta^*) = \arg \min_{\boldsymbol{\theta}, \boldsymbol{\alpha}, \beta} L(\boldsymbol{\theta}, \boldsymbol{\alpha}, \beta). \quad (5)$$

During this process, architecture parameters $\boldsymbol{\alpha}$ and $\beta$ are optimized into the best architecture suitable for the task, ultimately outputting the optimal multimodal object detection architecture.

### 3.2.2. EVOLUTIONARY ALGORITHM SEARCH STAGE

The second stage employs evolutionary algorithm optimization (Real et al., 2019). As shown in Figure 2, after several iterations, a set of high-performance candidate architectures is screened based on the differentiable weight distribution to serve as the initial population for the evolutionary stage. Assuming the initial population contains $N$ individuals, each individual $A_i$ is composed of a set of architecture parameters $\boldsymbol{\alpha}$ and $\beta$, i.e.,

$$A_i = [\boldsymbol{\alpha}_1, \boldsymbol{\alpha}_2, \ldots, \boldsymbol{\alpha}_k, \beta], \ (i = 1, \ldots, N), \quad (6)$$

where $\boldsymbol{\alpha}_j \in \{t_1, t_2, t_3, \ldots, t_m\}$ $(j = 1, \ldots, k)$, $m$ refers to the number of the multimodal symmetric fusion methods, $t_\tau$ is the $\tau$-th multimodal symmetric fusion methods $(\tau = 1, \ldots, m)$, $\boldsymbol{\alpha}_j$ represents the $j$-th operation be selected in the network. And the parameter $\beta$ indicates whether the current GFS module is enabled, i.e., the current GFS module is enabled when sigmoid$(\beta) \uparrow 1$, and disabled when sigmoid$(\beta) \downarrow 0$. Each individual architecture $A_i$ is trained on the multimodal object detection task, and its fitness is evaluated by calculating the objective loss function. The fitness function is typically based on detection accuracy (such as mAP) and the computational complexity of the model (such as FLOPs or parameter count). The fitness function is as follows:

$$f(A_i) = -(\lambda_{task} \cdot L_{task}(A_i) + \lambda_{com} \cdot L_{com}(A_i)), \quad (7)$$

where $L_{task}(A_i)$ is the loss of the multimodal object detection task (for example, the classification and regression losses), measuring the performance of the architecture. The complexity loss function $L_{com}$ is usually related to the computational load or parameter count of a network. The nonnegative parameters $\lambda_{task}$ and $\lambda_{com}$ are adjustment parameters that balance the trade-off between the performance of a

---

**Algorithm 1** Closed-loop Hybrid Search of MFH-NAS

**Require:** Visible-infrared dataset $\mathcal{D}$, backbone $\mathcal{B}$, fusion stages $n$, fusion operators $m$, alternating rounds $T$, retained candidates $K$.
**Ensure:** Optimal multimodal fusion detector $\mathcal{M}^\star$.
1: $\mathcal{S} \leftarrow \text{BuildSpace}(\mathcal{B}, n, m)$.
2: $\alpha_c, \beta_c \leftarrow \text{InitArch}(\mathcal{S}); \theta \leftarrow \text{InitWeights}(\mathcal{S})$.
3: **for** $t = 1$ to $T$ **do**
4: $\quad (\alpha_d, \beta_d, \theta) \leftarrow \text{DiffSearch}(\mathcal{S}, \alpha_c, \beta_c, \theta, \mathcal{D})$.
5: $\quad P_i \leftarrow \text{TopK}(\alpha_d, \beta_d, K)$.
6: $\quad P_{\text{par}} \leftarrow \text{Eval}(P_i, \mathcal{S}, \mathcal{D})$.
7: $\quad E \leftarrow \text{SelectElites}(P_{\text{par}})$.
8: $\quad P_c \leftarrow \text{Crossover}(E) \cup \text{Mutate}(E)$, s.t. $|P_c| = K$.
9: $\quad P_{\text{off}} \leftarrow \text{Eval}(P_c, \mathcal{S}, \mathcal{D})$.
10: $\quad C \leftarrow \text{SortByFitness}(P_{\text{par}} \cup P_{\text{off}})$.
11: $\quad A_{\text{best}}, F_{\text{best}} \leftarrow C[0]$.
12: $\quad \mathcal{I}_{op}, \mathcal{I}_{gate} \leftarrow \text{ActiveIdx}(A_{\text{best}})$.
13: $\quad \Delta\alpha \leftarrow \alpha_d[\mathcal{I}_{op}]/10; \Delta\beta \leftarrow \beta_d[\mathcal{I}_{gate}]/10$.
14: $\quad \alpha_c \leftarrow \alpha_d + \Delta\alpha; \beta_c \leftarrow \beta_d + \Delta\beta$.
15: **end for**
16: $(\bar{\alpha}, \bar{\beta}) \leftarrow \text{Discretize}(\alpha_c, \beta_c)$.
17: $\mathcal{M}^\star \leftarrow \text{Instantiate}(\mathcal{S}, \bar{\alpha}, \bar{\beta})$.
18: $\mathcal{M}^\star \leftarrow \text{FullTrain}(\mathcal{M}^\star, \mathcal{D})$.
19: **return** $\mathcal{M}^\star$.

---

task and the complexity of a model. In all main experiments, we set $\lambda_{\text{task}} = 1$ and $\lambda_{\text{com}} = 0.1$.

By (7), a higher fitness value indicates better performance on the target task. Subsequently, architectures with high fitness are selected as the parent population to evolve a new generation through selection, crossover, and mutation operations. The selection operation chooses high-performing individuals from the current population based on the fitness function. The crossover operation involves exchanging parts of genes (architectural parameters) between two parent individuals to produce new offspring. The mutation operation introduces small-scale random modifications to certain individuals' genes to enhance population diversity. Mutations can randomly select an operation at a certain layer within the architecture parameters for modification. The new generation continues to undergo fitness evaluation, selection, crossover, and mutation in the next iteration until the stopping criterion is met.

Algorithm 1 summarizes the closed-loop search process. MFH-NAS first constructs a fusion search space over candidate fusion stages and fusion operators, and initializes the continuous architecture parameters together with the network weights. At each alternating round, Stage I performs differentiable search in the supernet to update the continuous parameters, thereby learning local operator preferences and GFS activation tendencies. The resulting architecture distribution is then converted into the top-$K$ discrete candidates, which are evaluated and refined in Stage II through elite selection, crossover, mutation, and fitness evaluation. The best discrete architecture in the current round is used to iden-

tify the selected fusion operators and enabled GFS modules. Their corresponding architecture parameters are extracted to construct a feedback correction $(\Delta\alpha, \Delta\beta)$, which initializes the next differentiable round. This forms a closed loop between continuous operator optimization and discrete topology-level refinement. After the alternating process converges, the final architecture is obtained by retaining active GFS modules and selecting the highest-weight operator at each retained location.

## 4. Experiment

This section reports the experimental setup and empirical evaluation of MFH-NAS. We first describe the implementation details, datasets, and evaluation metrics. We then evaluate MFH-NAS under two search settings: (i) single-fusion-method stage search on M3FD to compare different stage-searching strategies, and (ii) multi-operator search on M3FD, LLVIP, and RGBT-Tiny to jointly optimize fusion stages and fusion operators. Finally, we conduct ablation studies on M3FD to quantify the contribution of each component, including differentiable search, evolutionary search, and the multi-operator search space.

### 4.1. Experimental Details

MFH-NAS is implemented using PyTorch (version 2.7.1) and trained across six NVIDIA RTX 5090 GPUs, leveraging CUDA 12.9. We evaluate the performance of MFH-NAS on three widely-used benchmark datasets: M3FD, LLVIP, and RGBT-Tiny. For each modality stream, YOLO11n is employed as the base detector backbone. During the architecture search process, we consider five stages of the backbone as potential fusion locations, in addition to input-level fusion, resulting in a total of six candidate fusion stages. We construct a set of fusion operators by selecting eight strong fusion methods from a pool of twelve recent RGB-T fusion strategies.

The search process is conducted over multiple stages. In Stage I, a differentiable architecture search is performed for 1000 epochs, with an initial learning rate of 0.025, a minimum learning rate of 0.001, and a learning rate of 3e-4 for architecture parameters. Every 100 epochs, a discrete candidate architecture is derived, and the top-performing candidates are used to initialize the population for Stage II. In Stage II, an evolutionary search is run for 10 generations, with the architecture learning rate set at 3e-4 and a Beta learning rate of 1e-3. We use the SGD optimizer with a momentum of 0.9, weight decay of 3e-4 for the network, 1e-3 for architecture parameters, and 1e-3 for Beta parameters. Gradient clipping is applied with a threshold of 5. The architecture search alternates between differentiable search and evolutionary search until the architeture search is complete. The final model is evaluated using standard object detection metrics, reporting the mean Average Precision (mAP) at a threshold of 0.5 (mAP@0.5) and the more stringent mAP averaged over IoU thresholds from 0.5 to 0.95 (mAP@0.5:0.95).

### 4.2. Comparative Analysis of Single Fusion Method Search

For systematic comparison, twelve RGB–T fusion methods proposed in recent years are selected as baselines, including: COMO (Liu et al., 2026), CMADet (Song et al., 2024), GM-DETR (Xiao et al., 2024), ICAFusion (Shen et al., 2024), CDC-YOLO (Wang et al., 2024b), DAMSDet (Guo et al., 2024), DHANet (Wu et al., 2025), M-SpecGene (Zhou et al., 2025), Fusion Mamba (Dong et al., 2025), UniRGB-IR (Yuan et al., 2024), TFDet (Zhang et al., 2024), and SCFR (Sun et al., 2024). In addition, four representative NAS-based methods, namely DC-NAS (Liang et al., 2024b), BM-NAS (Yin et al., 2022), EG-NAS (Cai et al., 2024), and SMoA (Liu et al., 2021), are included for comparison. For each fusion method, MFH-NAS is further compared with the four representative NAS approaches to investigate the effectiveness of different search strategies in exploring fusion stages on the M3FD dataset.

The experimental results are reported in Table 1, from which it can be observed that the fusion architectures discovered by MFH-NAS outperform, in most cases, both the other NAS-based methods and the corresponding handcrafted fusion baselines. Compared to handcrafted fusion methods such as COMO (0.826) and DAMSDet (0.802), our approach achieves a significantly higher average mAP50 of 0.842, demonstrating a clear improvement. When applied to the Fusion Mamba fusion method search, our approach reaches a peak mAP50 of 0.879, outperforming the original method by 4.5 percentage points. This demonstrates that the Gated Fusion Strategy (GFS) module can adaptively bridge the semantic gap between visible and infrared information, automatically exploring more optimal fusion stages. Additionally, in comparison with NAS models such as SMoA (0.841) and EG-NAS (0.840), MFH-NAS, through its dual-layer collaborative optimization mechanism, effectively mitigates the risk of being trapped in local optima—an issue commonly faced by pure gradient-based search methods—resulting in a more superior fusion detection architecture.

### 4.3. Comparative Analysis of Multi-Fusion Method Search

The stage-wise fusion configurations discovered by different NAS methods are summarized in Table 2. In the *In* column, the codes $\{0, 1, 2, 3\}$ indicate no input, visible-only input, infrared-only input, and dual-modality input, respectively. The *Cur.* column specifies the fusion operator applied to the current-stage visible and infrared features, whereas *Pre.*

*Table 1.* Comprehensive comparison of MFH-NAS with representative hand-crafted and NAS-based fusion models using mAP50 and mAP50-95 metrics on M3FD dataset.The best results are in **bold** and the second best are underlined.

| METHODS | ORIGINAL METHOD | | DC-NAS | | BM-NAS | | EG-NAS | | SMoA | | MFH-NAS(OURS) | |
|---|---|---|---|---|---|---|---|---|---|---|---|---|
| | MAP50 | MAP50-95 | MAP50 | MAP50-95 | MAP50 | MAP50-95 | MAP50 | MAP50-95 | MAP50 | MAP50-95 | MAP50 | MAP50-95 |
| COMO | 0.853 | 0.488 | 0.855 | 0.479 | 0.840 | 0.481 | 0.847 | 0.486 | 0.843 | 0.481 | 0.842 | 0.481 |
| CMADET | 0.822 | 0.455 | 0.819 | 0.530 | 0.832 | 0.599 | 0.815 | 0.527 | 0.820 | 0.527 | 0.822 | 0.531 |
| GM-DETR | 0.808 | 0.447 | 0.848 | 0.561 | 0.841 | 0.558 | 0.839 | 0.555 | 0.841 | 0.555 | 0.845 | 0.563 |
| ICAFUSION | 0.837 | 0.556 | 0.839 | 0.557 | 0.848 | 0.564 | 0.853 | 0.569 | 0.843 | 0.556 | 0.849 | 0.567 |
| CDC-YOLO | 0.839 | 0.555 | 0.853 | 0.561 | 0.842 | 0.562 | 0.861 | 0.566 | 0.856 | 0.565 | 0.872 | 0.573 |
| DAMSDET | 0.802 | 0.529 | 0.842 | 0.581 | 0.833 | 0.581 | 0.847 | 0.579 | 0.839 | 0.583 | 0.840 | 0.579 |
| DHANET | 0.835 | 0.584 | 0.827 | 0.589 | 0.830 | 0.581 | 0.827 | 0.489 | 0.837 | 0.585 | 0.831 | 0.588 |
| M-SPECGENE | 0.835 | 0.394 | 0.841 | 0.559 | 0.855 | 0.559 | 0.851 | 0.569 | 0.856 | 0.569 | 0.859 | 0.562 |
| FUSION MAMBA | 0.834 | 0.560 | 0.865 | 0.433 | 0.842 | 0.472 | 0.866 | 0.569 | 0.871 | 0.414 | 0.879 | 0.592 |
| UNIRGB-IR | 0.814 | 0.541 | 0.841 | 0.382 | 0.845 | 0.391 | 0.839 | 0.551 | 0.837 | 0.401 | 0.849 | 0.391 |
| TFDET | 0.812 | 0.529 | 0.807 | 0.568 | 0.821 | 0.577 | 0.811 | 0.519 | 0.827 | 0.579 | 0.813 | 0.576 |
| SCFR | 0.823 | 0.522 | 0.806 | 0.584 | 0.809 | 0.582 | 0.827 | 0.534 | 0.819 | 0.580 | 0.806 | 0.587 |
| AVERAGE | 0.826 | 0.513 | 0.837 | 0.532 | 0.837 | 0.542 | 0.840 | 0.543 | 0.841 | 0.533 | **0.842** | **0.549** |

*Table 2.* Stage-wise fusion configurations discovered by different NAS methods.

| Method | Stage 0 | | Stage 1 | | | Stage 2 | | | Stage 3 | | | Stage 4 | | | Stage 5 | | |
|---|---|---|---|---|---|---|---|---|---|---|---|---|---|---|---|---|---|
| | In | Cur. | In | Cur. | Pre. | In | Cur. | Pre. | In | Cur. | Pre. | In | Cur. | Pre. | In | Cur. | Pre. |
| DC-NAS | 0 | 0 | 3 | 11 | 0 | 3 | 2 | 5 | 2 | 14 | 11 | 2 | 14 | 4 | 3 | 2 | 4 |
| BM-NAS | 0 | 0 | 0 | 0 | 0 | 3 | 5 | 0 | 3 | 5 | 5 | 0 | 0 | 0 | 0 | 0 | 0 |
| EG-NAS | 3 | 2 | 3 | 11 | 11 | 3 | 2 | 5 | 3 | 11 | 11 | 3 | 5 | 4 | 3 | 2 | 4 |
| SMoA | 0 | 0 | 0 | 0 | 0 | 3 | 5 | 0 | 3 | 5 | 5 | 3 | 5 | 5 | 0 | 0 | 0 |
| Ours | 0 | 0 | 0 | 0 | 0 | 3 | 4 | 0 | 2 | 14 | 4 | 2 | 14 | 4 | 0 | 0 | 0 |

specifies the operator used to further fuse the current-stage fused feature with the fused feature propagated from the previous stage. Operator indices 1–12 correspond to the RGB–T fusion methods listed in Section 4.2, while indices 13 and 14 denote direct feature transfer from the infrared and visible branches, respectively.

MFH-NAS initiates fusion at Stage 2, where dual-modality features are fed into the GFS module and fused using ICA-Fusion. At Stages 3 and 4, the architecture selects infrared-only input and performs cross-stage fusion with the fused representation propagated from the previous stage, again using ICAFusion. The evolution of fusion-operator weights across all GFS modules during the search cycle is shown in Figure 5. To derive the final discrete architecture, we retain GFS modules with $\text{sigmoid}(\beta) > 0.5$ and select, for each retained module, the operator with the largest architecture weight $\alpha$. For comparison, DC-NAS applies SCFR at Stage 1 for current-stage fusion, uses CMADet at Stage 2, and then adopts CDC-YOLO for fusion with the previous-stage fused feature. At Stages 3 and 4, it selects infrared-only input and performs cross-stage fusion with SCFR and ICAFusion, respectively, and at Stage 5 it applies CMADet for current-stage fusion followed by ICAFusion for fusion with the previous fused feature.

We further evaluate multi-fusion-method search on M3FD, LLVIP, and RGBT-Tiny. Specifically, we rank the twelve

candidate fusion methods by mAP@0.5 on M3FD and select the top eight as fusion operators in the search space, namely (1) COMO, (2) CDC-YOLO, (3) ICAFusion, (4) DHANet, (5) M-SpecGene, (6) Fusion Mamba, (7) SCFR, and (8) CMADet. To examine whether the operator pool introduces target-benchmark bias, we further conduct an operator-pool transfer experiment. Instead of selecting candidate operators on the target benchmark M3FD, we first rank the twelve candidate fusion methods on LLVIP and select the top eight operators to form an LLVIP-selected pool. Compared with the original M3FD-selected pool, the LLVIP-selected pool removes SCFR and M-SpecGene and adds GM-DETR and DAMSDet. We then use this transferred operator pool to perform architecture search on M3FD. As shown in Table 4, the final performance remains almost unchanged: mAP@0.5 is identical, and mAP@0.5:0.95 differs by only 0.002. This suggests that the final architecture discovered by MFH-NAS is not tied to a particular target-benchmark-induced operator subset. We do not claim that MFH-NAS is fully benchmark-agnostic, but this result indicates that the method is substantially less sensitive to the source of operator pre-selection than the original concern might suggest. Under this setting, MFH-NAS is compared with four NAS baselines while jointly searching for the optimal combination of fusion stages and fusion operators. The results are summarized in Table 3.

On M3FD, YOLO11 achieves mAP@0.5 of 0.794 and 0.774

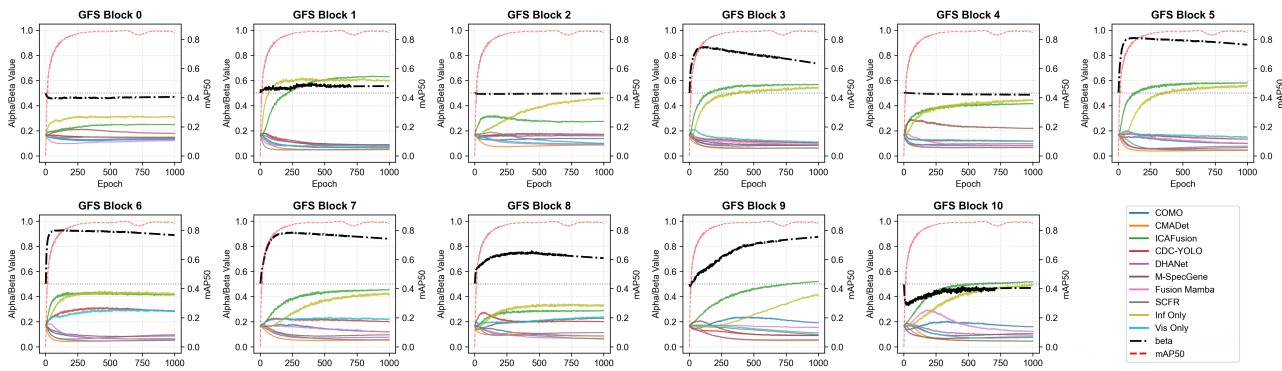

*Figure 5.* Evolution of Hyperparameters and Performance Analysis of Each Fusion Method in the GFS Module.

*Table 3.* Performance and search-cost comparison on M3FD, LLVIP, and RGBT-Tiny datasets. The best results are in **bold** and the second best are underlined. Search time is reported in GPU-days.

| METHODS | MODALITY | SEARCH TIME (GPU-DAYS) | M3FD | | LLVIP | | RGBT-TINY | |
|---|---|---|---|---|---|---|---|---|
| | | | MAP50 | MAP50-95 | MAP50 | MAP50-95 | MAP50 | MAP50-95 |
| YOLO11 | VIS | – | 0.794 | 0.519 | 0.951 | 0.611 | 0.433 | 0.212 |
| YOLO11 | IR | – | 0.774 | 0.505 | 0.958 | 0.628 | 0.464 | 0.230 |
| DC-NAS | VIS+IR | 2.8 | 0.869 | 0.522 | 0.954 | 0.632 | 0.474 | 0.242 |
| BM-NAS | VIS+IR | 0.9 | 0.877 | 0.531 | 0.959 | 0.632 | 0.469 | 0.239 |
| EG-NAS | VIS+IR | 1.8 | 0.852 | 0.489 | 0.956 | 0.654 | 0.472 | 0.242 |
| SMoA | VIS+IR | 1.1 | 0.871 | 0.567 | 0.958 | 0.649 | 0.465 | 0.243 |
| ParZC | VIS+IR | 1.5 | 0.873 | 0.531 | 0.957 | 0.660 | 0.472 | 0.239 |
| MFH-NAS | VIS+IR | 2.1 | **0.882** | **0.598** | **0.961** | **0.661** | **0.487** | **0.247** |

*Table 4.* Effect of operator-pool source on M3FD performance.

| Operator pool source | mAP@0.5 | mAP@0.5:0.95 |
|---|---|---|
| LLVIP-selected pool | 0.882 | 0.596 |
| M3FD-selected pool | 0.882 | 0.598 |

*Table 5.* Cross-dataset transfer of searched architectures without re-search.

| Eval \ Search | M3FD | LLVIP | RGBT-Tiny |
|---|---|---|---|
| M3FD | 0.882 | 0.864 | 0.755 |
| LLVIP | 0.922 | 0.961 | 0.761 |
| RGBT-Tiny | 0.311 | 0.296 | 0.487 |

in the RGB-only and IR-only settings, respectively, whereas MFH-NAS achieves 0.882, improving over the best single-modality baseline by 8.8 percentage points. MFH-NAS also attains the best mAP@0.5:0.95 of 0.598, exceeding the next best method (SMoA, 0.567) by 3.1 percentage points. On LLVIP, all methods perform strongly (mAP@0.5 > 0.95), and MFH-NAS achieves the highest mAP@0.5 of 0.961. On RGBT-Tiny, MFH-NAS reaches 0.487 mAP@0.5, outperforming the RGB-only YOLO11 baseline (0.433) and maintaining the best overall performance among NAS baselines in both mAP@0.5 and mAP@0.5:0.95. Qualitative detection results for architectures discovered by different NAS methods are visualized in Figure 6.

We further examine the robustness of discovered architec-

tures through cross-dataset transfer. Specifically, an architecture searched on one dataset is directly retrained and evaluated on another dataset without re-search. As shown in Table 5, each searched architecture performs best on its own dataset. For example, on M3FD, the M3FD-searched architecture reaches 0.882 mAP@0.5, while direct transfer from LLVIP- and RGBT-Tiny-searched architectures drops to 0.864 and 0.755, respectively. Similar trends can be observed on the other datasets. This indicates that RGB-T fusion architectures are dataset-adaptive rather than universal. Therefore, MFH-NAS should be interpreted as a search framework for discovering dataset-specific fusion architectures, instead of a single fixed architecture that transfers

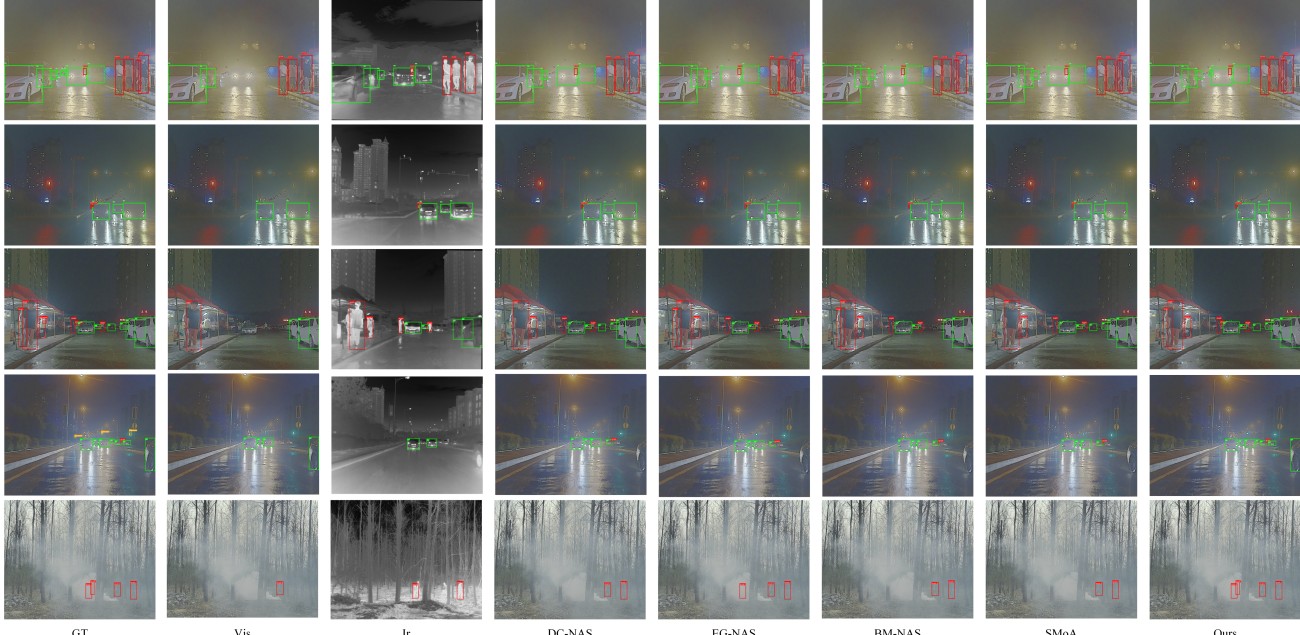

*Figure 6.* Visualization of Object Detection Results for Fusion Architectures by Different NAS Methods.

optimally across all RGB-T benchmarks.

### 4.4. Ablation Study

To validate the contributions of each core component in the proposed architecture, systematic ablation experiments were conducted on the M3FD dataset (Table 6). The experimental results show that relying solely on single-modality inputs (Index 1-2) leads to significant performance limitations. Upon introducing the dual-modality fusion mechanism and evolutionary search algorithm (Index 3), the mAP50 is significantly improved to 0.862. The incorporation of a differentiable search algorithm (Index 4) further boosts mAP50 to 0.872, demonstrating the ability of these search algorithms to more effectively locate optimal fusion paths within the complex architecture space. Subsequently, combining differentiable search with evolutionary search using a single fusion method (Index 5) results in a further increase in the mAP50-95 metric to 0.592. Finally, the complete search strategy employing multiple fusion methods (Index 6) achieves the optimal performance, with mAP50 and mAP50-95 reaching 0.882 and 0.598, respectively. This represents a significant improvement over the baseline fusion scheme, fully validating the superior effectiveness of combining evolutionary algorithms, differentiable search, and multiple fusion operators in optimizing the fusion architecture and enhancing object detection accuracy.

*Table 6.* Ablation study on the M3FD dataset to evaluate the effectiveness of search mechanisms and fusion strategies.

| MODALITY | | EA | DARTS | SINGLE METHOD | MULTIPLE METHODS | M3FD | |
| VIS | IR | | | | | MAP50 | MAP50-95 |
| --- | --- | --- | --- | --- | --- | --- | --- |
| ✓ | | | | | | 0.794 | 0.519 |
| | ✓ | | | | | 0.774 | 0.505 |
| ✓ | ✓ | ✓ | | | ✓ | 0.862 | 0.555 |
| ✓ | ✓ | | ✓ | | ✓ | 0.872 | 0.566 |
| ✓ | ✓ | ✓ | ✓ | ✓ | | 0.879 | 0.592 |
| ✓ | ✓ | ✓ | ✓ | | ✓ | **0.882** | **0.598** |

## 5. Conclusion

We presented MFH-NAS, a hybrid neural architecture search framework for multimodal fusion object detection that jointly optimizes fusion operators and stage-level connectivity to address modality gaps and enable cross-stage interactions. By coupling differentiable search for efficient local operator selection with evolutionary search for global topology exploration, MFH-NAS empirically alleviates early operator collapse and discovers high-performing fusion architectures with a reasonable accuracy-efficiency trade-off. Experiments on LLVIP, RGBT-Tiny, and M3FD show that MFH-NAS consistently outperforms handcrafted fusion-stage designs and prior stage-searching NAS baselines, improving mAP@0.5 from 85.3% to 88.2% over strong fixed-stage fusion methods on M3FD. These results suggest that jointly searching fusion primitives and cross-stage connectivity is a promising direction for robust RGB-T multimodal perception.

## Acknowledgements

This research was supported by the Scientific Startup Foundation for Doctors of Northwest A&F University (Z1090125002, Z1090324139).We thank the HPC platform of NWAFU for providing computational resources.

## Impact Statement

This paper presents work aimed at advancing the field of multimodal fusion in object detection. The research explores novel approaches to automatic architecture search for multimodal fusion, which could have significant implications for various applications such as autonomous driving and security surveillance. While the direct societal impact of this work is broad, it primarily contributes to improving the accuracy of multimodal systems, fostering advancements in both the academic and practical realms of artificial intelligence. There are no immediate ethical concerns that require specific attention.

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
