# OpenReview forum: "MFH-NAS:A Hybrid Neural Architecture Search Framework for Multimodal Fusion Object Detection"
_ICML.cc/2026/Conference — ICML 2026 regular_

### Official Review · Reviewer_HHhv · 2026-03-02

**Soundness:** 3
**Presentation:** 2
**Significance:** 3
**Originality:** 2
**Overall Recommendation:** 3
**Confidence:** 4

**Summary:**

This paper proposes  MFH-NAS, a hybrid neural architecture search framework for  Multimodal  Fusion Object Detection, which aims to automatically discovers fusion architectures to better leverage cross-modal complementarity. Specifically, MFH-NAS integrate gradient-based optimization and evolutionary search through a dual-level collaborative optimization mechanism. Specifically, a Gated Fusion Strategy (GFS) Selector is designed to select the multimodal features. Furthermore,
MFH-NAS  exploits the efficiency of differentiable search to rapidly optimize the weights of fusion operators at the microscopic level,  an evolutionary algorithm is employed to extensively explore the global search space at the macroscopic level. Experimental results show improvements on the LLVIP, RGBT-Tiny and M3FD datasets.

**Compliance With Llm Reviewing Policy:**

Affirmed.

**Final Justification:**

The rebuttal improves clarity and strengthens the empirical presentation.

 Since different fusion methods are known to be sensitive to backbone, neck, and training protocols, and re-implementing them under YOLO11n may alter their relative strengths.

In addition, the method relies on dataset-specific NAS, where architectures are optimized for particular benchmarks (e.g., M3FD, LLVIP, RGBT-Tiny). While this is common practice, it leaves open questions about how well the discovered architectures transfer across datasets, modalities, or detection frameworks.

Overall, the paper presents a solid engineering effort with reasonable empirical validation. However, the conceptual novelty and depth appear limited for ICML. I therefore maintain my original evaluation.

**Key Questions For Authors:**

- Novelty and Insight:
Beyond combining differentiable operator selection and evolutionary topology search, what is the core methodological novelty of MFH-NAS compared to prior hybrid NAS frameworks?

- Fairness of Operator Re-implementation:
Since the twelve RGB–T fusion methods are not their original implementation, how do the authors ensure that performance comparisons are fair and not biased by architectural or training differences?

- Search Cost and Efficiency:
What is the total search cost of MFH-NAS, and how does it compare to other NAS baselines under similar computational budgets? and

**Limitations:**

The authors should acknowledge that the method is validated only on RGB–Thermal detection and with a specific backbone (YOLO11n), which may limit generalization to other modalities or tasks. In addition, NAS methods can be computationally intensive.

**Strengths And Weaknesses:**

Strengths

- Problem setting. The paper explores NAS in the context of multispectral object detection, which is less studied before.

- Empirical validation across datasets. The method is evaluated on three public RGB–Thermal benchmarks (LLVIP, RGBT-Tiny, and M3FD), demonstrating consistent gains.

- Systematic search space design. The formulation of stage selection and operator selection is clearly structured, and the search space is well defined.

Major Weaknesses

- Limited methodological novelty. The core components such as differentiable operator selection and evolutionary topology search, are well-established techniques. The contribution largely lies in combining existing NAS components and applying them to multimodal fusion object detection. The conceptual and algorithmic novelty appears incremental rather than foundational.

- Limited baseline coverage. The comparisons are restricted to DC-NAS, BM-NAS, EG-NAS, and SMoA. It would strengthen the claims to include broader NAS baselines.

- Performance gains are relatively modest. Although the paper claims “strong improvements,” the margins on LLVIP, RGBT-Tiny, and M3FD are moderate in absolute terms.

- Lack of search cost comparison. The paper does not provide a clear comparison of search cost against other NAS methods. Given the hybrid strategy, computational efficiency is an important factor.

- Potential fairness issue in operator comparison. The twelve RGB–T fusion methods were originally optimized under their own backbone + neck + detection head configurations. In this work, they are implemented within a YOLO11n framework with modified training settings. This may not constitute a fully fair reproduction of the original methods and could affect the validity of direct performance comparisons.

Minor Weaknesses

- Scope of the title is broad. The title refers to “Multimodal Fusion Object Detection,” yet experiments are limited to RGB–Thermal detection. The claimed scope may be overstated.

- Missing computational complexity analysis. The paper would benefit from reporting model complexity metrics (e.g., parameters, FLOPs) for the searched architectures.

- Writing clarity. The manuscript would benefit from further polishing for clarity, conciseness, and precision in technical descriptions.

---

> ### Author Rebuttal · Authors · 2026-03-31
>
> Thank you for the careful review and constructive feedback. We appreciate your recognition of the problem setting, empirical validation, and search-space design.
>
> # R4-Q1: Novelty and insight
>
> We agree that MFH-NAS does not introduce a fundamentally new differentiable NAS or evolutionary NAS algorithm in isolation. Its main novelty lies in **formulating multimodal fusion detection as a joint search problem over both fusion-stage selection and fusion-operator selection**, and in coupling the two search levels through a dual-level optimization scheme. Compared with prior hybrid NAS frameworks, the key contributions are:
> (1) a search objective tailored to **non-aligned multimodal fusion**, jointly modeling **where to fuse** and **how to fuse**;
> (2) the **GFS selector**, which adaptively decides whether to fuse and which operator to apply at each stage;
> (3) a **closed-loop collaborative optimization process**, where Stage I learns local operator importance and Stage II explores global topology and feeds the best discrete structure back into the next round.
> Thus, the core novelty of MFH-NAS lies in its **unified formulation of multimodal fusion architecture search and its dual-level optimization design**. We will make this point clearer in the revision. We also agree that the gains are moderate in absolute terms, and we will report them more precisely on a per-dataset basis.
>
> # R4-Q2: Fairness of operator re-implementation
>
> We acknowledge that the 12 RGB-T fusion methods are not fully faithful reproductions of their original implementations, since all of them are re-implemented under a **YOLO11n** framework whose backbone, neck, and detection head differ from the original works. Our goal is therefore not to reproduce the absolute numbers reported in the original papers, but to provide a **relatively fair comparison** of fusion strategies under a **unified and controlled setting**. We use the same detection framework, dataset splits, training schedule, and hyperparameters for all compared methods, so the resulting differences mainly reflect the fusion strategies themselves. We will clarify that these results should be interpreted as **relative comparisons under a unified framework**, not as direct reproductions of the original methods.
>
> # R4-Q3: Search cost and efficiency
>
> We will add a search-cost comparison in **Table 3**, reported in **GPU-days**; for reference, **1 GPU-days = 24 GPU-hours**.
>
> |Method|Search time (GPU-days)|
> |:--:|:--:|
> |DC-NAS|2.8|
> |BM-NAS|0.9|
> |EG-NAS|1.8|
> |SMoA|1.1|
> |ParZC|1.5|
> |**MFH-NAS(ours)**|**2.1**|
>
> Under our setting, MFH-NAS alternates two stages: Stage I runs **1000 epochs** of differentiable search, and after every **100 epochs**, the current candidates are passed to Stage II for **10 epochs** of evolutionary search, whose result is fed back to the next round. The total search cost is **2.1 GPU-days**. This is lower than purely evolutionary NAS such as **DC-NAS**, but higher than lighter differentiable NAS such as **BM-NAS**. We view this as a reasonable efficiency–performance trade-off.
>
> # R4-Q4: Broader NAS baseline coverage
>
> |method|mAP@0.5|mAP@0.5:0.95|
> |:--:|:--:|:--:|
> |ParZC|0.873|0.531|
> |MFH-NAS|**0.882**|**0.594**|
>
> We agree that broader NAS baseline coverage would strengthen the empirical claims. We therefore additionally include **ParZC (AAAI 2025)** as a new baseline. Its search cost is **1.5 GPU-days**, and the resulting architecture achieves **mAP@0.5 = 0.873** and **mAP@0.5:0.95 = 0.531**. We will add it to Table 3 and update the comparison accordingly. While this does not exhaust all possible NAS baselines, it broadens the empirical coverage within the rebuttal timeline.
>
> # R4-Q5: Computational complexity analysis
>
> We agree that complexity metrics are important for evaluating the practical value of the searched architectures. We will therefore report both the **number of parameters** and **GFLOPs** in the revised manuscript. The collected results are:
>
> |Method|Parameters(M)|GFLOPs|
> |:--:|:--:|:--:|
> |DC-NAS|13.12|8.0|
> |BM-NAS|12.47|7.6|
> |EG-NAS|13.55|8.0|
> |SMoA|13.36|7.4|
> |ParZC|12.74|8.1|
> |MFH-NAS(ours)|12.78|7.6|
>
> These results show that, under the current setting, MFH-NAS achieves the best detection performance with competitive complexity, rather than relying on a noticeably larger model. We will add this table and discuss the complexity–performance trade-off in the revised manuscript. We will update Table 2 in the revised manuscript accordingly.
>
> # Acknowledgment
>
> We also agree with the scope-related concerns. The current validation is limited to **RGB–Thermal detection** with **YOLO11n**, so generalization to other modalities or tasks remains to be studied. NAS methods can also be computationally intensive. We will make these limitations more explicit, slightly narrow the title/scope wording, and further polish notation, pseudocode, and technical descriptions for clarity and reproducibility.

---

> > ### Author Rebuttal · Reviewer_HHhv · 2026-04-04
> >
> > Thank you for the response. The rebuttal addresses several concerns. But it remains unclear how the proposed formulation substantially differs from existing hybrid NAS paradigms.
> >
> > In addition, the use of a unified YOLO-based framework provides controlled conditions, but does not fully ensure fairness. I think additional evidence is needed to demonstrate that the re-implementations are representative and unbiased.

---

> > > ### Author Response · Authors · 2026-04-08
> > >
> > > ## Q1. Novelty and insight
> > >
> > > Thank you for the follow-up. We agree that MFH-NAS is **not** a new generic hybrid NAS optimizer. Its contribution is a **task-specific formulation for multimodal fusion object detection** with topology-aware hybrid search. As summarized below, the difference from prior hybrid NAS such as **EG-NAS** is mainly in the **search space, search object, and coupling mechanism**.
> > >
> > > | Aspect  | EG-NAS (AAAI 2024) | MFH-NAS (this work)           |
> > > | --| ---| -- |
> > > | **Target task**             | Generic image classification NAS on DARTS / NAS-Bench-201 search spaces | Multimodal fusion object detection for RGB-T benchmarks      |
> > > | **Search space**            | Standard DARTS-style **single-modal cell-level search space**, mainly over candidate operations and connections | **Task-specific multimodal fusion search space**, explicitly including: (i) fusion-stage selection and (ii) fusion-operator selection |
> > > | **Hybrid search mode**      | Mostly a **one-way cooperation**: gradient descent updates network weights, while evolutionary strategy explores search directions for architecture updates | **Two-stage closed-loop alternating optimization**: differentiable search first learns local fusion preferences, then evolutionary search explores global fusion topology, and the best structure is fed back to the next differentiable round |
> > > | **Architecture parameters** | Primarily architecture parameter **\(\alpha\)**, updated using ES-generated directions | Jointly searches **\(\alpha\)** (fusion operator selection) and **\(\beta\)** (GFS activation / skipping decisions) |
> > > | **Search object**           | Generic operation-level architecture search                  | Joint search of **where to fuse** and **how to fuse**, including stage-level fusion and cross-stage propagated fusion interactions |
> > > | **Main contribution**       | Improving search-direction exploration in generic hybrid NAS | Introducing a **task-specific search formulation for non-aligned multimodal fusion**, together with a topology-aware hybrid optimization mechanism |
> > >
> > > The key distinction is therefore the **problem formulation**: EG-NAS remains in a generic single-modal cell space, while MFH-NAS explicitly searches **where to fuse** and **how to fuse** for multimodal fusion detection, using both \(\alpha\) and \(\beta\) and an explicit alternating feedback loop between differentiable and evolutionary search. Our novelty lies **less in a new generic optimizer** and **more in a multimodal-fusion-specific search formulation with stronger topology-aware coupling**. We will revise the paper to make this scope clearer.
> > >
> > > ## Q2. Fairness of operator re-implementation
> > >
> > > Thank you for the follow-up. We understand that the remaining concern is whether the unified YOLO-based re-implementations are sufficiently **representative and unbiased**. To test this directly, we performed an additional validation experiment: we first reproduced the 12 fusion methods from their public implementations, and then migrated the same fusion modules into the same **YOLO11n-based unified framework** under the same dataset, backbone, training schedule, and hyperparameter settings.
> > >
> > > ### (1) Results under the original open-source implementations
> > >
> > > |Method|mAP@0.5|
> > > |---|---:|
> > > |FusionMamba|0.879|
> > > |COMO|0.851|
> > > |CDC-YOLO|0.839|
> > > |M-SpecGene|0.836|
> > > |DHANet|0.835|
> > > |ICAFusion|0.831|
> > > |CMADet|0.826|
> > > |SCFR|0.821|
> > > |TFDet|0.814|
> > > |GM-DETR|0.814|
> > > |UniRGB-IR|0.811|
> > > |DAMSDet|0.805|
> > >
> > > ### (2) Results after migrating the same fusion modules into the unified YOLO11n framework
> > >
> > > |Method|mAP@0.5|
> > > |--|--:|
> > > |COMO|0.853|
> > > |CDC-YOLO|0.839|
> > > |ICAFusion|0.837|
> > > |M-SpecGene|0.835|
> > > |DHANet|0.835|
> > > |FusionMamba|0.834|
> > > |SCFR|0.823|
> > > |CMADet|0.822|
> > > |UniRGB-IR|0.814|
> > > |TFDet|0.812|
> > > |GM-DETR|0.808|
> > > |DAMSDet|0.802|
> > >
> > > After migration to YOLO11n, **absolute performance and some local rankings do change**, which is expected. The key result is that the **overall relative trend is preserved**: the **top-8 candidate operators are exactly the same in the two settings (8/8 overlap)**, and the **Spearman rank correlation is approximately 0.84**. Therefore, the unified YOLO11n setting is **not** a perfectly faithful reproduction of the original methods, but it is a **controlled and reasonably representative comparison protocol** under a common detector and training setup. Under this interpretation, the additional experiment shows that migration to YOLO11n does not materially distort the overall strength ordering of the candidate fusion methods or the operator pool used in our search space. We will revise the paper to state this more explicitly and to clarify that these results should be interpreted as **a controlled plug-in comparison under a unified YOLO11n detector**, rather than as a leaderboard-style reproduction of the original papers.

---

### Official Review · Reviewer_jntx · 2026-03-02

**Soundness:** 3
**Presentation:** 3
**Significance:** 3
**Originality:** 2
**Overall Recommendation:** 4
**Confidence:** 4

**Summary:**

The paper proposes MFH‑NAS, a hybrid neural architecture search framework for multimodal fusion in object detection. It combines gradient‑based differentiable search  with evolutionary algorithm search through a dual‑level collaborative optimization mechanism. A Gated Fusion Strategy (GFS) Selector is introduced to adaptively decide whether to fuse features at each stage and which fusion operator to apply. The method is evaluated on three RGB‑T datasets and compared against several hand‑crafted fusion methods and existing NAS baselines. Results show consistent improvements, demonstrating the effectiveness of the proposed hybrid search strategy.

**Compliance With Llm Reviewing Policy:**

Affirmed.

**Final Justification:**

The paper presents a solid hybrid search framework with clear empirical improvements, and I find the overall approach and motivation reasonable. My initial concerns were mainly about the clarity of the β mechanism, search cost transparency, operator selection rationale, and generalization across datasets. The rebuttal addresses these points well: it clarifies the role and optimization of β with more concrete descriptions and pseudocode, provides explicit comparisons of search cost, justifies the operator selection process, and adds cross-dataset transfer results to better characterize generalization. I also appreciate the authors’ acknowledgment that the work is currently stronger empirically than theoretically and their plan to improve clarity in the revision. While the theoretical grounding and broader generalization remain somewhat limited, the rebuttal significantly improves the clarity and completeness of the work. Overall, these updates positively change my assessment, and I recommend a weak accept.

**Key Questions For Authors:**

1.What is the total GPU cost (hours or FLOPs) of the entire MFHNAS search process, and how does it compare to the baseline NAS methods (DCNAS, BMNAS, etc.)?

2.How exactly is the β parameter optimized during the differentiable search stage? Is it jointly trained with the architecture parameters α, and how is it represented and evolved in the evolutionary stage?

3.Why were these eight specific fusion operators selected from the twelve recent methods? Was any criterion (e.g., performance, diversity) used to prune the set?

4.Have you tested the architectures discovered on M3FD directly on LLVIP or RGBTTiny without further search? If so, how well do they perform?

**Limitations:**

Yes

**Strengths And Weaknesses:**

Strengths

1.The hybrid search framework effectively combines the fast convergence of differentiable NAS with the global exploration of evolutionary algorithms, mitigating the localoptima problem common in pure gradientbased fusion search.

2.The Gated Fusion Strategy (GFS) module provides a learnable, stagewise mechanism to decide whether and how to fuse multimodal features, increasing architectural flexibility.

3.The analysis of discovered fusion configurations (Table 2) reveals interesting patterns (e.g., skipping earlystage fusion) that offer insights for future multimodal detector design.

Weaknesses

1.The optimization details of the GFS control parameter β are unclear: how β is updated during the differentiable stage and how it is encoded in the evolutionary population are not specified.

2.The set of fusion operators is simply selected from eight recent methods without justification for why those eight are chosen or whether the search could discover operators beyond this fixed set.

3.No cross‑dataset generalization experiments are conducted; it is unknown whether the fusion architectures found on one dataset (e.g., M3FD) transfer well to others without re‑search.

---

> ### Author Rebuttal · Authors · 2026-03-31
>
> Thank you for the review. We appreciate your recognition of the hybrid search framework and GFS.
> # R3-Q1: Total search cost
> MFH-NAS costs **2.1 GPU-days** (about **50.4 GPU-hours**), lower than **DC-NAS (2.8)** but higher than **BM-NAS (0.9)**, **SMoA (1.1)**, and **EG-NAS (1.8)**. All costs are reported under our re-implementation setting to support consistent relative comparison. We will report this comparison explicitly in the revision; a broader table is also given in **R4-Q3**. We will update Table 3 in the revised manuscript accordingly.
> # R3-Q2: Clarification of the β mechanism
> We agree that the optimization and representation of **β** are not stated clearly enough. In **Stage I**, β is jointly optimized with α as a continuous architecture parameter. In **Stage II**, β is represented by the **activation state of each GFS gate**.
>
> **Input**: dataset $\mathcal{D}$, backbone $\mathcal{B}$, stages $n$, operators $m$, rounds $T$, retained candidates $\mathrm{top}\text{-}K$
> **Output**: detector $\mathcal{M}^\star$
>
> 1. $\mathcal{S} \leftarrow \mathrm{BuildFusionSearchSpace}(\mathcal{B}, n, m)$
> 2. $\alpha_{\mathrm{cur}}, \beta_{\mathrm{cur}} \leftarrow \mathrm{InitializeArchParams}(\mathcal{S});\ \theta \leftarrow \mathrm{InitializeNetworkWeights}(\mathcal{S})$
> 3. **FOR** $t=1$ **TO** $T$ **DO**
> 4. $\quad (\alpha_{\mathrm{diff}}, \beta_{\mathrm{diff}}, \theta) \leftarrow \mathrm{DifferentiableSearch}(\mathcal{S}, \alpha_{\mathrm{cur}}, \beta_{\mathrm{cur}}, \theta, \mathcal{D})$
> 5. $\quad P_{\mathrm{init}} \leftarrow \mathrm{SelectTopKCandidates}(\alpha_{\mathrm{diff}}, \beta_{\mathrm{diff}}, \mathrm{top}\text{-}K)$
> 6. $\quad \mathrm{ParentSet} \leftarrow \mathrm{EvaluatePopulation}(P_{\mathrm{init}}, \mathcal{S}, \mathcal{D})$
> 7. $\quad \mathrm{EliteSet} \leftarrow \mathrm{SelectElites}(\mathrm{ParentSet})$
> 8. $\quad P_{\mathrm{child}} \leftarrow \mathrm{Crossover}(\mathrm{EliteSet}) \cup \mathrm{Mutate}(\mathrm{EliteSet}), \quad \text{s.t. } |P_{\mathrm{child}}| = \mathrm{top}\text{-}K$
> 9. $\quad \mathrm{OffspringSet} \leftarrow \mathrm{EvaluatePopulation}(P_{\mathrm{child}}, \mathcal{S}, \mathcal{D})$
> 10. $\quad \mathrm{CandidateSet} \leftarrow \mathrm{SortByFitness}(\mathrm{ParentSet} \cup \mathrm{OffspringSet})$
> 11. $\quad A_{\mathrm{best}}, f_{\mathrm{best}} \leftarrow \mathrm{CandidateSet}[0]$
> 12. $\quad I_{\mathrm{op}}, I_{\mathrm{gate}} \leftarrow \mathrm{ExtractActiveIndices}(A_{\mathrm{best}})$
> 13. $\quad \Delta \alpha \leftarrow \alpha_{\mathrm{diff}}[I_{\mathrm{op}}]/10,\quad \Delta \beta \leftarrow \beta_{\mathrm{diff}}[I_{\mathrm{gate}}]/10$
> 14. $\quad \alpha_{\mathrm{cur}} \leftarrow \alpha_{\mathrm{diff}} + \Delta \alpha,\quad \beta_{\mathrm{cur}} \leftarrow \beta_{\mathrm{diff}} + \Delta \beta$
> 15. **END FOR**
> 16. $(\bar{\alpha}, \bar{\beta}) \leftarrow \mathrm{DiscretizeArchParams}(\alpha_{\mathrm{cur}}, \beta_{\mathrm{cur}})$
> 17. $\mathcal{M}^\star \leftarrow \mathrm{InstantiateModel}(\mathcal{S}, \bar{\alpha}, \bar{\beta})$
> 18. $\mathcal{M}^\star \leftarrow \mathrm{FullTrain}(\mathcal{M}^\star, \mathcal{D})$
> 19. **RETURN** $\mathcal{M}^\star$
>
> As shown above, **β is jointly trained with α in Stage I** and represented by **GFS gate states** in Stage II. After each evolutionary round, the active gate indices from the best discrete architecture are used to extract the corresponding entries in $\beta_{\mathrm{diff}}$, forming $\Delta \beta$ for the next round. In the final discretization, only modules satisfying **sigmoid(β) > 0.5** are retained. We will make this mechanism more explicit in the revised manuscript.
> # R3-Q3: Justification of operator selection
>
> Specifically, under a **unified YOLO11n setting**, we first evaluated 12 candidate fusion methods and retained the top 8 mainly based on **preliminary performance and stability** under the same training protocol, while operator diversity was secondary. The goal was to remove ineffective or unstable operators and keep the search tractable. **Operator diversity was not the primary pruning criterion**, and the search is restricted to a **fixed operator pool**. Discovering new operators beyond this set is outside the present scope. We will clarify this in the revised manuscript.
>
> # R3-Q4: Cross-dataset generalization
>
> We added a **cross-dataset transfer experiment**: an architecture searched on one dataset is directly retrained/evaluated on another **without further search**. Each searched architecture performs best on its own dataset, while direct transfer causes a drop. This indicates **dataset-adaptive fusion architectures** rather than a universal one; results are provided in **R1-Q1c**.
>
> # Brief note
> We agree that the current paper is stronger empirically than theoretically. We will clarify key hyperparameters, pseudocode, search cost, and workflow, and moderate wording about search dynamics and global exploration.

---

> > ### Author Rebuttal · Reviewer_jntx · 2026-04-03
> >
> > The authors have addressed my concerns, but after considering everything, I will maintain my weak accept recommendation.

---

### Official Review · Reviewer_SXy7 · 2026-03-12

**Soundness:** 3
**Presentation:** 4
**Significance:** 3
**Originality:** 4
**Overall Recommendation:** 5
**Confidence:** 3

**Summary:**

The paper introduces MFH-NAS, a hybrid neural architecture search framework specifically designed for multimodal fusion object detection. Authors propose a system that automatically discovers optimal fusion architectures by searching both local fusion primitives and global stage-level connectivity. The main contribution is a dual-level optimization strategy: a differentiable search efficiently learns architecture parameters for local fusion operations, while an evolutionary search explores global topologies to prevent the system from getting trapped in local optima.

**Compliance With Llm Reviewing Policy:**

Affirmed.

**Final Justification:**

The authors provided a highly detailed rebuttal that successfully tackled my primary questions. I commend their effort, as their clear explanations regarding the experimental setup resolved my initial reservations. This constructive dialogue reinforces my prior assessment of the paper's impact and clarity, leading me to confidently maintain my initial score.

**Key Questions For Authors:**

1-How sensitive is the evolutionary search phase (Stage II) to the specific initialization provided by the differentiable search (Stage I)? Have you run control experiments using random initializations for the evolutionary stage to quantify the exact benefit of Stage I?
2-The fitness function for the evolutionary search relies on λtask and λcom to balance task performance and model complexity. Could you clarify the exact numerical values used for these hyperparameters during your experiments, and how sensitive the final architecture is to these specific weights?
3-In Table 3, the RGBT-Tiny baseline performance for the IR-only modality is notably low (0.230 mAP@0.5) compared to the VIS-only modality (0.433 mAP@0.5). Does the GFS selector dynamically learn to suppress the weaker/noisier modality entirely in these specific edge cases?

**Limitations:**

yes

**Strengths And Weaknesses:**

For soundness, submission is technically sound and methodologically rigorous. Authors successfully identify the limitations of purely gradient-based differentiable search methods and logically introduce an evolutionary algorithm to ensure global exploration. The experimental design is comprehensive, testing the framework across three public benchmarks. However, from an analytical perspective, while the empirical validation is strong, the paper relies strongly on these experimental results rather than providing theoretical bounds or proofs regarding the convergence of their specific dual-level alternating optimization loop.
For presentation, I think the manuscript is clearly written, the overall narrative is structured well and easy to follow. The authors provide helpful visuals, such as the overview of the four-stage pipeline, which effectively guides the reader through the complex search space. One minor weakness might be that certain hyperparameters crucial for reproducing the evolutionary fitness function are discussed conceptually but could benefit from more explicit mathematical grounding in the text.
For significance, I believe that by introducing MFH-NAS, authors advance the capabilities of Neural Architecture Search in this domain, offering a novel hybrid approach that explores both local fusion primitives and global cross-stage connectivity. This contribution may provide substantial practical utility, as evidenced by consistent performance gains across multiple benchmarks, including M3FD, LLVIP, and RGBT-Tiny.
For originality, from my perspective, this manuscript demonstrates strong originality through a creative combination of existing techniques. While differentiable architecture search and evolutionary algorithms are established concepts, coupling them specifically to navigate the exponentially growing combinatorial complexity of non-aligned multimodal fusion provides a novel perspective.

---

> ### Author Rebuttal · Authors · 2026-03-31
>
> Thank you for the careful review and positive assessment. We greatly appreciate your recognition of the paper’s method design, experimental completeness, clarity of presentation, and originality. We respond to your key questions below and will incorporate the corresponding experiments, analyses, and clarifications in the revised manuscript.
>
> ## R2-Q1: Sensitivity of Stage II to Stage I initialization
>
> We added a controlled experiment comparing two Stage II initialization strategies: (i) **Top-$K$** candidate architectures obtained from Stage I, and (ii) **random initialization** within the same search space.
>
> | Stage II initialization          |  mAP@0.5  | mAP@0.5:0.95 |
> | :------------------------------- | :-------: | :----------: |
> | Stage I Top-\(K\) initialization | **0.882** |  **0.594**   |
> | Random initialization            |   0.864   |    0.575     |
>
> Using Stage I candidates yields gains of **0.018** on mAP@0.5 and **0.023** on mAP@0.5:0.95. This shows that Stage I provides informative rather than incidental initialization for Stage II. We also observed faster early convergence and smaller fluctuations. We will add this control experiment and discussion in the revision.
>
> ## R2-Q2: Values and sensitivity of $\lambda_{task}$ and $\lambda_{com}$
>
> In our experiments, we fix **$\lambda_{task}=1$** and conduct a sensitivity analysis on **$\lambda_{com}$**:
>
> | $\lambda_{task}$ | $\lambda_{com}$ | mAP@0.5 |
> | :----------------: | :---------------: | :-----: |
> |         1          |       0.01        |  0.881  |
> |         1          |       0.05        |  0.882  |
> |         1          |       0.10        |  0.882  |
> |         1          |       0.20        |  0.882  |
> |         1          |       0.30        |  0.876  |
> |         1          |       0.40        |  0.875  |
> |         1          |       0.50        |  0.879  |
>
> The method is not very sensitive to **$\lambda_{com}$**: when **$\lambda_{com}\in[0.01,0.5]$**, mAP@0.5 remains within **0.875–0.882**, with a maximum gap of **0.007**. The best performance (**0.882**) is achieved when **$\lambda_{com}\le 0.2$**, while performance decreases slightly when **$\lambda_{com}\ge 0.3$**, indicating that overly strong complexity constraints may mildly limit model capacity. We use **$\lambda_{task}=1$** and **$\lambda_{com}=0.1$** in the main experiments, and we will state these default settings explicitly in the fitness-function definition.
>
> ## R2-Q3: Modality imbalance and whether GFS suppresses weaker modalities
>
> We first clarify the metric indexing in **Table 3**: on **RGBT-Tiny**, **0.433** corresponds to YOLO11 **VIS-only mAP@0.5**, whereas **0.230** corresponds to **IR-only mAP@0.5:0.95**; the corresponding **IR-only mAP@0.5** is **0.464**. Therefore, the table does not indicate that IR-only mAP@0.5 is much lower than VIS-only mAP@0.5.
>
> Regarding the GFS behavior, the answer is **yes, but adaptively and stage-dependently**, rather than by fully disabling one modality everywhere. GFS uses the gating parameter **$\beta$** to decide whether a fusion module is active, and the final discrete architecture retains only modules with **$\mathrm{sigmoid}(\beta)>0.5$**. Under modality imbalance, some fusion modules related to the weaker modality learn smaller $\beta$ values, suppressing low-quality fusion, while higher-level cross-modal fusion may still be preserved when complementary information is useful. We will clarify the metric indexing in Table 3 and add this explanation in the revised manuscript.
>
> ## Brief note
>
> We agree with your observation that the current paper is **stronger empirically than theoretically**. We do not claim a formal convergence proof for the current dual-level loop and will phrase the related claims empirically. We also agree that some hyperparameters of the evolutionary fitness function are currently described too conceptually. In the revision, we will make the mathematical formulation and default settings more explicit, including
> $$
> f(A_i) = -\left(\lambda_{\text{task}}L_{\text{task}}(A_i) + \lambda_{\text{com}}L_{\text{com}}(A_i)\right),
> $$
> with $\lambda_{\text{task}}=1$ and $\lambda_{\text{com}}=0.1$, and we will state the main evolutionary hyperparameters more clearly in the text and pseudocode. We will clarify the key hyperparameters, pseudocode, and workflow, and moderate wording related to search dynamics and global exploration.

---

> > ### Author Rebuttal · Reviewer_SXy7 · 2026-04-04
> >
> > Thank you to the authors for the thorough, constructive, and highly transparent rebuttal. My concerns have been fully resolved.

---

### Official Review · Reviewer_JDLy · 2026-03-12

**Soundness:** 3
**Presentation:** 2
**Significance:** 2
**Originality:** 2
**Overall Recommendation:** 4
**Confidence:** 2

**Summary:**

This paper proposes MFH-NAS, a hybrid neural architecture search framework for multimodal fusion object detection, with the goal of jointly searching fusion operators and stage-level fusion connectivity rather than relying on predefined stage-aligned fusion designs. The method combines a differentiable search stage for local operator weighting and gating with an evolutionary search stage for global topology exploration, and alternates between the two to improve both efficiency and robustness. The paper evaluates the approach on M3FD, LLVIP, and RGBT-Tiny, and reports that the searched architectures outperform handcrafted fusion-stage baselines and several NAS-based alternatives. The authors proceed to address an important context, namely the underexplored problem of how to search cross-stage multimodal fusion architectures instead of only designing better fusion modules. The authors proceed to study a central domain in multimodal perception, namely RGB-T object detection under heterogeneous feature hierarchies and cross-stage semantic gaps. The overall message is that jointly searching fusion stage selection and fusion operator assignment can produce stronger multimodal detectors than fixed-stage fusion pipelines.

**Compliance With Llm Reviewing Policy:**

Affirmed.

**Final Justification:**

The authors have almost addressed my previous concerns. However, after reviewing comments from other reviewers, I still hold suspects to this method in terms of generalization and feasibility in practical. So I would raise my previous rating to borderline accept.

**Key Questions For Authors:**

1. Final accuracy alone is not sufficient to establish this point. The paper currently lacks convincing analysis of convergence behavior, search trajectories, stability across runs, and robustness of the discovered architectures.
2. Important components of the method remain underspecified, including the alternation strategy between the two search stages, candidate transfer rules, evolutionary hyperparameters, and stopping conditions. In its current form, the method is not described with enough precision for reproducibility.
3. There appears to be a contradiction between the definition of the fitness function and the textual description that higher fitness implies better performance. This is not a minor presentation issue, since it directly affects the correctness of the evolutionary optimization objective.
4. This design makes the search space partially benchmark-dependent, which raises a serious concern about comparison fairness and possible dataset bias. Please justify this choice and show whether the method still holds under a benchmark-agnostic operator pool.
5. The paper does not report repeated runs, variance, or sensitivity to random seeds. Given the stochastic nature of NAS, this omission makes it difficult to judge whether the observed improvements are statistically reliable.

**Limitations:**

This work has several limitations. First, although the method is empirically effective, the claims about improved global exploration and reduced premature convergence are only partially validated and remain more empirical than rigorously demonstrated. Second, the NAS setting appears computationally intensive, but the paper does not quantify search efficiency in enough detail to assess practical scalability. Third, parts of the search design depend on preselecting strong fusion methods based on benchmark performance, which may reduce the generality of the conclusion. Fourth, the experimental evidence focuses on three RGB-T detection benchmarks and does not yet establish robustness under broader domain shifts, alternative detectors, or different multimodal tasks. Finally, the presentation and notation still need refinement to make the method fully reproducible and easier to verify independently.

**Strengths And Weaknesses:**

Strengths:
The paper tackles a meaningful and timely problem. Much prior RGB-T detection work focuses on designing fusion modules while keeping the fusion topology largely fixed. This paper explicitly targets the architectural question of where to fuse and how to connect stages across modalities, which is a worthwhile direction and is reasonably motivated in the introduction and method sections.
A second strength is the overall methodological framing. The combination of differentiable search and evolutionary search is sensible: the former offers efficient local optimization over candidate fusion primitives, while the latter is intended to improve global exploration over stage-level connectivity. Conceptually, this hybridization is more compelling than using only a pure DARTS-style or pure EA-style search, especially in a large combinatorial multimodal search space. The paper also provides a reasonably clear search-space description, including a gated fusion selector and an estimate of the resulting architecture-space size.
The empirical results are broadly positive. On the reported benchmarks, MFH-NAS achieves the best performance among the compared NAS methods, including on M3FD and RGBT-Tiny, and shows gains over both single-modality baselines and several searched fusion baselines. The ablation study also suggests that combining differentiable search, evolutionary search, and multiple fusion methods is better than using only a subset of these ingredients.
Weaknesses:
1.Several technical claims in the paper are stronger than the evidence actually supports. The paper argues that the hybrid search alleviates premature convergence and improves global exploration, but the supporting evidence is mostly indirect, mainly final performance results. It provides little direct analysis of search dynamics, convergence behavior, run-to-run stability, or the robustness of the discovered architectures. Although Figure 5 visualizes weight evolution, it is still insufficient to convincingly show that the hybrid process effectively avoids local optima.
2.The search method is not described in sufficient detail for reproducibility. Although the differentiable stage is presented in a DARTS-like manner, several key practical details remain unclear.
3.The paper also contains some inconsistencies and ambiguities in its formulation and presentation. For example, Equation (7) defines the fitness function using task loss and complexity loss, while the text states that a higher fitness indicates better performance.
4.The paper is original, but its contribution is more of a meaningful incremental improvement than a fundamental breakthrough. Its core idea is to combine two established NAS paradigms on top of a task-specific search space and a gated fusion selector.
5.Although the experimental section is relatively comprehensive, the fairness of some comparisons remains unclear. The paper first selects twelve RGB-T fusion methods as baselines, then ranks them according to their performance on M3FD and retains the top eight as candidates for the multi-operator search space. This choice is not necessarily problematic, but it means that the search space itself is partially shaped by performance on a specific benchmark, and thus should be discussed more carefully, as it may introduce dataset bias into the results.

---

> ### Author Rebuttal · Authors · 2026-03-31
>
> Thank you for the careful review. We appreciate your recognition of our problem setting, hybrid search design, and empirical results.
>
> # R1-Q1: Multi-dimensional Analysis
>
> We agree that final accuracy alone is insufficient. We therefore add analyses of **convergence**, **search trajectories**, and **architecture robustness**; **run-to-run stability** is addressed in **R1-Q5**.
>
> ## R1-Q1a: Convergence
>
> Thank you for the suggestion. We add a direct comparison of the convergence behavior of pure differentiable search and MFH-NAS. Under pure differentiable search, **10 of 11 GFS modules converge within the first 50 epochs**, and the architecture parameters rapidly collapse to a single dominant operator, showing clear **early convergence and insufficient exploration**. In contrast, under MFH-NAS, about **5 modules do not stabilize until after 200 epochs**, while **multi-operator competition** is preserved much longer. This provides more direct empirical evidence that the hybrid search strategy reduces early operator collapse. We will add the corresponding curves and discussion in the revision.
>
> ## R1-Q1b: Search trajectories
>
> As shown in Fig. 5, the search exhibits a clear stage-wise pattern: during **0–100 epochs**, operator weights separate rapidly; during **100–400 epochs**, a small subset gradually becomes dominant; after **400 epochs**, the weights largely stabilize. Each GFS module typically ends with only **1–2 dominant operators** (\(\alpha \approx 0.4\)–\(0.6\)), suggesting relatively stable structure selection under the current setting. We will add this analysis in the revision.
>
> ## R1-Q1c: Architecture robustness
>
> We add a **cross-dataset transfer experiment**, where an architecture searched on one dataset is directly retrained/evaluated on another **without re-search**:
>
> |Eval \ Search|M3FD|LLVIP|RGBT-Tiny|
> | :--: | :--: | :--: | :---: |
> |M3FD| **0.882** |0.864|0.755|
> |LLVIP|0.922| **0.961** |0.761|
> |RGBT-Tiny|0.311|0.296| **0.487**|
>
> Each searched architecture performs best on its own dataset. On **M3FD**, the M3FD-searched architecture reaches **0.882**, while direct transfer from LLVIP- and RGBT-Tiny-searched architectures drops to **0.864** and **0.755**. Similar trends hold on the other datasets, indicating **dataset-adaptive fusion architectures**. We will include this experiment and discussion in the revision.
>
> # R1-Q2: Reproducibility
>
> We agree that the alternating optimization is not described explicitly enough. In the revision, we will clarify the alternation frequency, the Top-$K$ transfer rule, the feedback mechanism, and the main evolutionary hyperparameters: **population size 20**, **mutation rate 0.3**, **uniform crossover** with **0.5** per-position parent selection, **50% elitism**, and termination at the **maximum number of generations**. The updated pseudocode is provided in **R3-Q2**. To further improve reproducibility, we also plan to release the code upon acceptance.
>
> # R1-Q3: Fitness definition
>
> We acknowledge this issue and revise Eq. (7) as
> $$
> f(A_i) = -\left(\lambda_{\text{task}} L_{\text{task}}(A_i) + \lambda_{\text{com}} L_{\text{com}}(A_i)\right).
> $$
> Higher fitness then consistently corresponds to better individuals during evolutionary selection. We will update the text accordingly.
>
> # R1-Q4: Operator-pool fairness
>
> We agree that a benchmark-agnostic operator-pool check is important. We have therefore started an additional experiment that selects candidate operators on **LLVIP** and uses them for architecture search on **M3FD**. This directly tests whether MFH-NAS remains effective when the operator pool is not shaped by the target benchmark. Since the experiment is computationally expensive (about **20 GPU-days**), it has not yet finished within the rebuttal period, and we therefore keep this as an explicit limitation.
>
> # R1-Q5: Repeated runs and seed sensitivity
>
> We conducted **5 independent runs**:
>
> |Seed|mAP@0.5|mAP@0.5:0.95|
> |:--:|:--:|:--:|
> |1|0.883|0.596|
> |2|0.879|0.582|
> |3|0.883|0.598|
> |4|0.882|0.607|
> |5|0.881|0.587|
> |**Mean±Std**|**0.882±0.0017**|**0.594±0.0098**|
>
> The mean results are very close to the originally reported **0.882 / 0.598**, with only **0.0004** drop in mAP@0.5 and **0.004** drop in mAP@0.5:0.95. This indicates good stability across seeds. The searched architectures are also largely consistent: **ICAFusion is always selected as the key operator, and the main fusion stages remain around Stages 2–4**. Due to time and computational budget constraints, we have not repeated all baseline NAS methods under multiple seeds, but these results help verify that our gain is not due to a favorable single run.
>
> # Acknowledgment
> The concerns about incremental novelty and search cost are addressed in **R4-Q1** and **R3-Q1**, respectively. The current experiments focus on RGB-T detection with a specific backbone, while extending the study to broader domain shifts, alternative detectors, and other multimodal tasks is an important direction for future work.

---

> > ### Author Rebuttal · Reviewer_JDLy · 2026-04-02
> >
> > The authors have almost addressed my previous concerns. However, after reviewing comments from other reviewers, I still hold suspects to this method in terms of generalization and feasibility in practical. So I would raise my previous rating to borderline accept.

---

> > > ### Author Response · Authors · 2026-04-08
> > >
> > > Thank you for reading our rebuttal carefully and for raising your assessment to **borderline accept**. We especially appreciate your remaining concern regarding **generalization** and **practical feasibility**. To further address your **R1-Q4** concern about a **benchmark-dependent operator pool** and the resulting possibility of dataset bias, we have now completed the additional experiment that was still ongoing during the rebuttal period.
> > >
> > > Specifically, under the same **unified YOLO11n protocol**, we first evaluated all 12 candidate fusion methods on **LLVIP** and obtained the following results:
> > >
> > > | Method | mAP@0.5 | mAP@0.5:0.95 |
> > > |---|---:|---:|
> > > | CDC-YOLO | 0.979 | 0.693 |
> > > | DAMSDet | 0.979 | 0.791 |
> > > | DHANet | 0.977 | 0.702 |
> > > | GM-DETR | 0.974 | 0.702 |
> > > | CMADet | 0.971 | 0.701 |
> > > | ICAFusion | 0.971 | 0.691 |
> > > | COMO | 0.970 | 0.771 |
> > > | FusionMamba | 0.970 | 0.643 |
> > > | SCFR | 0.964 | 0.702 |
> > > | M-SpecGene | 0.963 | 0.741 |
> > > | TFDet | 0.961 | 0.711 |
> > > | UniRGB-IR | 0.961 | 0.722 |
> > >
> > > We then ranked these methods by **mAP@0.5**, selected the top 8 to form a new operator pool, and used this **LLVIP-selected pool** for architecture search on **M3FD**. Compared with the original **M3FD-selected** pool, the LLVIP-selected pool removes **SCFR** and **M-SpecGene** and adds **GM-DETR** and **DAMSDet**. The final performance on **M3FD** is:
> > >
> > > | Method | mAP@0.5 | mAP@0.5:0.95 |
> > > |---|---:|---:|
> > > | MFH-NAS (operators selected on LLVIP) | 0.882 | 0.596 |
> > > | MFH-NAS (operators selected on M3FD) | 0.882 | 0.598 |
> > >
> > > The key observation is that, when the candidate operator pool is no longer selected on the target benchmark itself but instead transferred from another benchmark (**LLVIP**), the final performance remains almost unchanged: **the mAP@0.5 is identical, and the mAP@0.5:0.95 differs by only 0.002**. This suggests that, although pre-screening does change the operator set, the final architecture discovered by MFH-NAS is **highly robust to the source of the operator pool**, and the observed gains are **not tied to a particular target-benchmark-induced operator subset**.
> > >
> > > We will add this experiment to the revised manuscript as a direct follow-up to **R1-Q4** and phrase the conclusion more carefully. We do **not** claim that MFH-NAS is fully benchmark-agnostic; however, this result provides direct empirical evidence that the method is substantially less sensitive to the benchmark used for operator pre-selection than the original concern might suggest. We believe this new result also helps alleviate the remaining concerns about **generalization** and **practical feasibility**. Thank you again for your careful review and constructive feedback.

---

### Decision · Program_Chairs · 2026-04-30

**Decision:**

Accept (regular)

**Comment:**

Four reviewers gave overall positive scores: one ‘Accept’, two ‘Weak accept’, and one ‘Weak reject’. During the rebuttal phase, the authors successfully addressed almost all the concerns raised by the reviewers, except for the one regarding novelty (Reviewer HHhv). Based on all of these, the decision is to recommend the paper for acceptance. However, the authors should clarify the novelty in the camera-ready version if the paper is finally accepted.